# Challenges of Teleneurology in the Care of Complex Neurodegenerative Disorders: The Case of Parkinson’s Disease with Possible Solutions

**DOI:** 10.3390/healthcare11243187

**Published:** 2023-12-17

**Authors:** Seyed-Mohammad Fereshtehnejad, Johan Lökk

**Affiliations:** 1Edmond J. Safra Program in Parkinsonߣs Disease and Morton and Gloria Shulman Movement Disorders Clinic, Toronto Western Hospital, UHN, Toronto, ON M5T 2S8, Canada; 2Institute of Health Policy, Management and Evaluation (IHPME), Dalla Lana School of Public Health, University of Toronto, Toronto, ON M5S 1A1, Canada; 3Division of Clinical Geriatrics, Department of Neurobiology, Care Sciences and Society (NVS), Karolinska Institutet, 171 77 Stockholm, Sweden; johan.lokk@regionstockholm.se

**Keywords:** telemedicine, teleneurology, neurodegeneration, Parkinson’s disease, virtual encounter

## Abstract

Teleneurology is a specialist field within the realm of telemedicine, which is dedicated to delivering neurological care and consultations through virtual encounters. Teleneurology has been successfully used in acute care (e.g., stroke) and outpatient evaluation for chronic neurological conditions such as epilepsy and headaches. However, for some neurologic entities like Parkinson’s disease, in which an in-depth physical examination by palpating muscles and performing neurologic maneuvers is the mainstay of monitoring the effects of medication, the yield and feasibility of a virtual encounter are low. Therefore, in this prospective review, we discuss two promising teleneurology approaches and propose adjustments to enhance the value of virtual encounters by improving the validity of neurological examination: ‘hybrid teleneurology’, which involves revising the workflow of virtual encounters; and ‘artificial intelligence (AI)-assisted teleneurology’, namely the use of biosensors and wearables and data processing using AI.

## 1. Introduction 

Neurological diseases are among the primary causes of disability worldwide, and they present a steadily increasing burden, as measured by disability-adjusted life-years (DALY) [1]. The increase in the all-age prevalence and burden of neurodegenerative entities such as Alzheimer’s dementia and Parkinson’s disease (PD) are particularly evident in developed and in many developing countries, mostly due to population aging and enhanced longevity [2]. On the other hand, there is a growing gap between the supply of and demand for neurologists, with a severe shortage of general neurologists and subspecialists in movement disorders, which is more pronounced in low-income nations but is also easily recognizable in high-income countries [3,4]. According to the World Health Organization (WHO), there is a substantial and growing imbalance in the distribution of neurologists, with many regions facing a severe shortage. For instance, there are only three adult neurologists per 10 million people in low- and middle-income countries, whereas high-income countries have around 160 times more neurologists in proportion to their population [5]. This disparity leaves vast populations with limited access to specialized neurological care. Although it is of a lower magnitude, the same problem also exists in most high-income countries, fueled by factors such as an aging population and the rising prevalence of neurological disorders. According to the American Academy of Neurology (AAN), by 2025, it is projected that 41 states in the United States will labor under a mismatch between the need for neurologists and the availability of neurologists, and 88% of these states will experience a mismatch of over 20% [6]. One major consequence of this imbalance between demand and supply is the long waiting time for patients with neurodegenerative conditions to be seen in movement-disorder or memory clinics. For instance, a study performed before the recent pandemic in Hamilton, Canada, estimated the mean wait time for neurologic assessment to be around 110 days [7]. A survey conducted in England, including over 10,000 patients, revealed that individuals with neurological disorders such as PD encounter significant delays in accessing specialized healthcare services; the wait time can be over 12 months after referral by a general practitioner in one-third of these patients [8]. The wait times for subspecialty care visits have been remarkably prolonged during and after the SARS-CoV-2 pandemic. In practice, wait times have been extended to as long as 12 months for the assessment of patients suspected of having PD or other Parkinsonian syndromes referred to movement-disorder clinics in some countries. Long wait times have been shown to negatively affect patient outcomes, prohibit timely medical care, and restrict access to non-emergency care [9]. In the case of people with PD, delayed diagnosis and treatment can deteriorate quality of life, worsen mobility, and increase the risk of falls. 

The emergence of telemedicine as a health informatics solution using digital telecommunication tools dates to more than two decades ago, with early expansion around the 2000s to help with remote access to healthcare and reduce wait times. As the mainstay of telemedicine, virtual care refers to healthcare appointments that are held by using distant methods, such as video conferences or remote patient monitoring. The use of virtual care has grown exponentially since the SARS-CoV-2 pandemic to align with social distancing guidelines [10]. Virtual care is probably the only efficient option with which to provide neurological care to an aging population with a rapidly growing burden of neurodegenerative diseases. However, important challenges and shortcomings remain in the use of telemedicine and virtual clinical encounters. One key challenge is the reliance on comprehensive neurological examinations, which traditionally involve hands-on assessments of motor and sensory functions, as well as cerebellar, gait, and cognitive changes. The limitations of virtual platforms in capturing these nuances can hinder accurate diagnosis and disease monitoring. A recent scoping review of 37 studies on teleneurology highlighted the essential complementary role of traditional in-person healthcare in conjunction with telemedicine to increase the value of virtual care for neurological diseases [11]. Additionally, the elderly population affected by Alzheimerߣs or other forms of dementia (including Parkinsonߣs disease dementia) may face barriers related to technology literacy and access, posing challenges for their effective engagement in telehealth initiatives [12]. The lack of standardized protocols for remote neurological assessments and the potential for variations in the quality of internet connectivity further contribute to the challenges in ensuring consistent and reliable teleneurology services. Accelerated attempts during the SARS-CoV-2 pandemic have resulted in the development of guidelines for virtual neurological examinations [13], which should be considered to improve the quality of neurological examinations in teleneurology. As technology evolves and healthcare systems adapt, addressing these challenges will be essential for optimizing the efficacy of teleneurology in delivering comprehensive and patient-centered care for those with neurodegenerative diseases. 

In this manuscript, we aim to further explore an important caveat in telemedicine: the low validity and feasibility of physical examination, with particular implications for PD. Two potential health informatics solutions are recommended and critiqued to tackle this problem.

## 2. Methods

To select relevant articles, we performed a comprehensive exploration of the literature on the PUBMED/MEDLINE database. The primary objective was to identify publications, including original studies and review articles, focusing on teleneurology applications in the context of neurodegenerative diseases. The search utilized a combination of keywords and Medical Subject Heading (MeSH) terms, including ‘teleneurology’, ‘telemedicine’, ‘neurodegenerative diseases’, ‘Parkinson’s disease’, ‘Alzheimer’s disease’, and related terms. The inclusion criteria were restricted to articles published in English, aligning with the language of the review. A systematic approach was adopted, incorporating Boolean operators to refine the search scope and ensure relevance to the objective of assessing teleneurology interventions in the context of neurodegenerative conditions. This strategy aimed to capture a comprehensive and representative body of research to inform the narrative synthesis and discussions within the current prospective article.

## 3. Teleneurology: Current Situation and Challenges 

Teleneurology is a subspecialty of telemedicine that focuses specifically on providing neurological care and consultations remotely using telecommunications technology. In practice, teleneurology consists of virtual encounters in which patients and their neurologists are connected remotely via a secure telemedicine platform. Patients need a computer, smartphone, or tablet with a camera, microphone, and internet connection. Some platforms might require specific software or apps, which patients need to download and install in advance. At the scheduled time, patients and neurologists log in to the telemedicine platform. History taking, virtual physical examination, discussion, and care planning are all performed through a webcam, and neurologists e-prescribe medications and submit the necessary referrals and requisites for investigations (e.g., imaging, laboratory tests) through the electronic health record (EHR) system. 

In recent decades, teleneurology has been used successfully in both acute care settings, such as stroke (‘telestroke’) [14], and outpatient teleconsultations for chronic neurologic conditions [15], such as movement disorders [16], epilepsy [17], multiple sclerosis [18], dementia [19], and headache [20]. During and after the SARS-CoV-2 pandemic, there was a tremendous rise in the use of teleneurology; for instance, the rate increased from 3.9% to 94.6% at the Neurological Institute at Cleveland Clinic, USA, within a short time frame of only 3 weeks, in March 2020 [21]. The potential for this rapid adaptation relies on factors such as the readiness of technological infrastructure, isolation mandates related to the SARS-CoV-2 pandemic, and health-information literacy of staff. 

Studies have shown feasibility and equal effectiveness of teleneurology in some settings (e.g., stroke management via telestroke), as well as enhanced patient satisfaction and, in some cases, cost savings [22]. However, there are caveats with virtual encounters, especially in a specialty like neurology, in which proper in-person physical examination remains key in diagnosis, monitoring of treatment effects, and clinical decision making. Even with recent advances in diagnostic imaging and biomarkers, the field of neurology heavily relies on detailed history taking and physical examination of various neurological functions in peripheral and central nervous systems [23]. For individuals with PD, in the initial consultation and at every follow-up visit, neurologists need to examine axial and appendicular tone of arms and legs for signs of rigidity by passively moving patients’ limbs, as well as testing for bradykinesia (slowness of motion) with repetitive movements, tremor, and gait evaluation. Parts of the examination can be performed virtually by demonstrating the tasks or through observation over the camera. However, it is impossible to test for muscle tone without physically palpating patients, and it is difficult to evaluate postural instability without an in-person pull test. In one study on the feasibility of various components of neurological examination in a virtual encounter, cranial nerve II/III examination was categorized as the low-feasibility group, while motor bulk/tone was in the medium-feasibility group, and the rest of the neurological examination was in the high-feasibility group [24]. In the field of neurosurgery, it has been shown that approximately 18.5% of virtual visits are unsuccessful, mainly due to insufficient physical examination, resulting in repeated in-person appointments for further assessment [25]. The need to repeat an appointment as in-person, if rises in number, can increase the cost to and burden on the healthcare system. 

In addition, teleneurology and telemedicine in general are facing other challenges. Ensuring robust security measures on telemedicine platforms is paramount for safeguarding patient privacy and data integrity. As the healthcare landscape increasingly embraces digital technologies, the confidential and sensitive nature of neurologic consultations mandates a heightened focus on security. A breach in telemedicine security, which can be associated with environmental, technological, and operational factors, not only jeopardizes patient trust but also exposes individuals to the risk of unauthorized access to their health information [26].

## 4. Hybrid Teleneurology: Revisited Virtual Care Workflow 

Even with its shortcomings, the use of teleneurology is projected to expand and become even more integrated into many facets of neurological practice. More recently, there have been attempts to optimize the quality and feasibility of outpatient virtual neurology clinics. One example is the development of the ‘Virtual Rapid Access Clinic’, involving multidisciplinary staff and online administrative tools to expedite patients’ access to care and maintain quality by providing an integrated multidisciplinary care model [27], which is indeed crucial in the context of PD [28]. The implementation of the ‘Virtual Rapid Access Clinic’ has shortened wait times by an average of 15 days and increased monthly patient-throughput rates [27]. 

One novel solution to the inefficiency of virtual physical examination is ‘hybrid teleneurology’, for which the workflow of virtual care needs to be revised. In the ‘gatekeeper model’ of the healthcare system, the routine workflow used to care for patients suspected of having a neurodegenerative disorder such as PD starts with a referral from the primary care provider (e.g., the family physician) to the neurologist. In routine teleneurology, the patient is seen later by a neurologist via a virtual encounter, with the previously mentioned barriers for a proper neurological examination. Family physicians, however, are usually more familiar with patients, are more geographically accessible, and can feasibly arrange in-person visits. Recent data suggest that even after a successful telemedicine experience during the pandemic, 80% of primary care physicians would prefer to conduct future patient visits in person rather than via telemedicine; this opinion is also held by patients themselves [29]. 

As illustrated in Figure 1, this workflow can be revised in the form of ‘hybrid teleneurology’, in which a neurologist is virtually connected to an in-person encounter at the family physician’s office, where a patient with a suspected neurodegenerative disease, like PD, is present. The family physician, as a trained and qualified healthcare worker, can perform a proper neurological examination (e.g., test muscle tone and muscle-stretch reflexes with a hammer, etc.) and even specific neurologic maneuvers, if guided by a virtual neurologist via a live connection. In addition to the improved validity of neurological examination, another advantage of this hybrid model of telemedicine is the possibility of faster and more efficient shared clinical decision making between the patient, primary care provider (e.g., family physician), and specialist (neurologist), all at once, in a single encounter. The hybrid encounters merge two separate visits into one and, consequently, save costs and shorten wait times for patients. This type of hybrid model is quite novel for outpatient encounters; nevertheless, another type of hybrid visit has been proposed for remote inpatient settings with poor access to neurologists, where a nurse or nurse practitioner is present at the bedside and a neurologist is present virtually, at the webside, to guide proper neurological assessments for hospitalized patients [4]. Another example of a hybrid teleneurology platform is the telestroke encounter, in which an emergency physician connects to a stroke neurologist via webcam for the virtual assessment of patients suspected of having acute stroke to guide neurologic examinations and acute management. It should be noted that the term ‘hybrid teleneurology’ has also been used in a different context, in which it refers to a model of care that combines both conventional in-person visits and remote virtual care. In one example of this type of ‘hybrid teleneurology’, the data from a neuromuscular clinic in Greece revealed equal quality of care during the SARS-CoV-2 pandemic between the ‘hybrid’ model, with both tele-consultation and face-to-face, in-person visits, and the conventional all-in-person visits provided during the pre-pandemic era [30]. The workflow is, however, quite different from our proposed model of ‘hybrid teleneurology.’ In traditional ‘hybrid telehealth’ models, patients interact with their physicians in two different ways: a typical virtual encounter through a camera and a conventional in-person visit to the clinic, occurring either before or after the virtual encounter. In this workflow, the potential risk of an invalid neurological exam during the virtual meeting still exists. Nonetheless, in our innovative proposed model of ‘hybrid teleneurology’, the virtual encounter with the neurologist is designed to occur simultaneously with the in-person visit by the patient to the primary physician (e.g., family physician), so that medically trained personnel can perform a proper neurological examination guided by a neurologist online. 

This solution can be criticized for presenting a higher administrative workload due to the arrangement of such hybrid visits, for which both the family physician (in person) and the neurologist (virtually) need to be available. Moreover, parts of the neurologic examination, such as testing for muscle tone and rigidity, are, at least to some extent, subjective, meaning that the inter-rater reliability can be low between different examiners (particularly between a neurologist and a non-neurologist) [31]. Therefore, neurologists might prefer to examine patients themselves in order to have a better sense of the severity of the symptoms to be monitored over time and to adjust the doses of medications during follow-ups.

## 5. Artificial Intelligence (AI)-Assisted Teleneurology: Role of Biosensors 

Smartphone apps, wearables, and biosensors, as well as machine-learning-powered augmented-reality systems, are enhancing the capabilities of telemedicine, including teleneurology [32]. The utilization of sensor-based remote monitoring has the potential to enhance the monitoring of PD progression and to evaluate the efficacy of potential disease-modifying pharmaceutical therapies. A recent study revealed moderate-to-strong correlations between a smartwatch-based app’s unsupervised remote measurements and those of the Movement Disorder Society-Sponsored Revision of the Unified Parkinson’s Disease Rating Scale (MDS-UPDRS), a standardized scale used to assess resting tremor, bradykinesia, and gait during in-person visits [33]. A large list of sensor-based wearable devices, including body-attached sensors, ubiquitous networking, and embedded sensors, has been applied to track motor symptoms; these tools are valuable in capturing baseline and longitudinal neurological data in people with PD [34]. For instance, using an inertial-measurement-unit-based motion-capture system, researchers objectively measured full-body tremor in a cohort of individuals with PD [35]. Lightweight and wearable inertial sensors have been proposed for monitoring the freezing of gait in people with PD [36]. Furthermore, many other e-health wearables have the ability to integrate contextual data [34]. Some examples are the DynaPort Hybrid system (worn on the lower back) [37], the Parkinson’s Kinetigraph (a wrist-worn logger) [38], the SENSE-PARK system (for monitoring gait, hypokinesia, dyskinesia, and sleeping) [39], and the GaitRite System (for monitoring gait parameters) [40], to name a few. Although the data collected from these biosensors can be noisy and sometimes difficult to interpret, these methods of continuous data collection have some strengths. Assessments conducted in clinics frequently fail to accurately represent the normal conditions within patients’ home environments; however, the wearables and sensors that are worn for long hours at home collect real-time data in real environments. 

As illustrated in Figure 2, in the AI-assisted-teleneurology model, validated wearables and biosensors can be used to enhance the value of neurological examinations during virtual encounters, as well as to remotely monitor patients’ motor and non-motor manifestations. By using AI, the self-tracking big data can be converted into meaningful outputs to assist neurologists in clinical decision making (e.g., changes or adjustments to the timing and dosage of medication). Previously, AI-assisted teleneurology was conceptualized in other contexts, such as telestroke, including the use of machine learning algorithms for the timely analysis of brain imaging in the acute phase of stroke, the provision of decision-support systems to assist with treatment plans (e.g., thrombolysis, thrombectomy), and speech-recognition tools, all of which can improve stroke outcomes [41]. 

There is a growing surge in the availability of these technologies worldwide, and many patients already possess technologies that can supplement the remote neurologic examination [42]. However, there are critiques and concerns to consider. A significant number of patients with neurodegenerative conditions such as PD are old and, therefore, potentially not technologically literate; furthermore, they may develop cognitive impairments even during the prodromal stages [43], resulting in challenges in the appropriate use of smartphones or biosensors [44]. Another critique of the use of wearables and biosensors is the risk of a low signal-to-noise ratio and an increased cognitive burden for physicians. The potential for bias in AI algorithms poses challenges, as it can lead to disparities in diagnosis and treatment recommendations [45]. Most importantly, the integration of AI in teleneurology, and in healthcare in general [46], necessitates a careful consideration of ethical implications. Ethical concerns surround issues such as data privacy, security, and patient consent, given the sensitive nature of neurological health information. Continuous data monitoring in healthcare, while offering valuable insights, raises legitimate privacy concerns that necessitate the careful consideration of patient consent. Patients may express concerns about the extent to which their data are monitored, shared, or stored, prompting the need for transparent and comprehensive consent processes. Adequate informed-consent procedures must be in place, clearly outlining the scope of the data’s collection, their potential uses, and any sharing practices. Transparency in AI decision-making processes and engaging patients in this process is crucial to building trust among patients and healthcare professionals. Additionally, issues related to accountability, responsibility, and the potential for the over-reliance on AI recommendations merit careful consideration. To ensure privacy or security measures, robust encryption protocols are required to secure data transmission during remote consultations. Access controls and authentication mechanisms together with regular security audits should be implemented to restrict unauthorized access to AI systems and patient databases. The ongoing development of teleneurology presents a significant challenge in maintaining ethical norms while effectively utilizing AI to improve diagnostic accuracy and patient care in people with neurodegenerative disorders. The continuous consideration of ethical principles and the development of explicit protocols are important to effectively negotiate the intricacies involved and guarantee the conscientious implementation of AI technologies within the field of neurological healthcare.

## 6. Future Directions

The future of teleneurology holds immense promise in the integration of wearable devices for data collection and the utilization of AI technologies. Wearable devices equipped with advanced sensors can offer continuous, real-time monitoring of neurological parameters, providing invaluable insights into patients’ daily activities and health trends. This opens avenues for more comprehensive and personalized care, especially in managing chronic neurological conditions. The continuous streams of data from wearables present rich opportunities for research into disease progression, treatment efficacy, and the identification of early warning signs. Furthermore, the application of AI in teleneurology can significantly enhance diagnostic accuracy, treatment planning, and patient management. Machine learning algorithms can be used to analyze complex neurological data patterns, aiding in the early detection of abnormalities and in predicting disease trajectories, particularly in the case of neurodegenerative diseases. 

The future of ‘hybrid teleneurology’ lies in innovative shifts in workflow that aim to elevate the quality of data collection and enhance the overall efficiency of remote neurological assessments. Advancements in user-friendly interfaces, coupled with intuitive data-capture methods, will streamline the process of information exchange between patients and neurologists. The emphasis will be on refining the components of standard neurological examinations to suit remote settings, fostering a more dynamic and interactive approach. The aim of this evolution in workflows is to overcome barriers associated with traditional telehealth encounters, ensuring that ‘hybrid teleneurology’ not only remains a viable alternative but emerges as a preferred and effective mode of delivering neurological care. Collaborative efforts between primary care physicians, neurologists, IT technologists, and data scientists will be essential in unlocking the full potential of teleneurology, ushering in a new era of patient-centered, data-driven neurological care. As technology continues to advance, the synergy between wearable devices and AI holds the key to transformative breakthroughs in remote neurological healthcare.

## 7. Conclusions

While teleneurology offers unprecedented opportunities for remote patient care, our review underscores the challenges associated with virtual neurological examinations, particularly in the context of diseases like PD. In the era of teleneurology, in-person physical examinations must be supplemented by virtual versions as neurologists transition from traditional to virtual workflows. However, virtual examinations are limited in value, and they are not feasible for some neurologic maneuvers, such as testing for rigidity in PD. Recognizing the potential for innovation, two novel models were proposed in this article to enhance the yield of virtual encounters by focusing on improving the value of neurological examinations: ‘hybrid teleneurology’, which involves revising the workflows of virtual encounters; and ‘AI-assisted teleneurology’, which is focused on applying information technology tools and data processing using AI. Similar technologies and models might be applied in some centers; however, we proposed innovative modifications to these teleneurology models to improve their efficacy, particularly the value of virtual neurological examination. The ‘hybrid teleneurology’ model redefines the workflows of virtual encounters by seamlessly integrating in-person neurological exams at the office of the referring family physician while the neurologist is simultaneously connected online to guide the examination and live consultation. This approach preserves crucial hands-on assessments while leveraging the convenience of telehealth. Furthermore, the ‘AI-assisted teleneurology’ model introduces a paradigm shift by harnessing information technology tools and AI-driven data processing on information gathered from wearables and biosensors. This not only enhances the diagnostic accuracy of virtual assessments but also enables the continuous monitoring and early detection of neurological changes. Many biosensors and wearables provide the capability of integrating contextual data by continuously quantifying motor and, to a lesser extent, some non-motor features of PD, with excellent potential to be embedded in current ‘teleneurology’ clinics. As we navigate the evolving landscape of teleneurology, these innovative models hold the potential to revolutionize patient care, bridging the gap between remote consultations and comprehensive neurological examinations. Embracing these models will pave the way for a more inclusive, effective, and patient-centric approach to neurology care in the digital age. 

## Figures and Tables

**Figure 1 healthcare-11-03187-f001:**
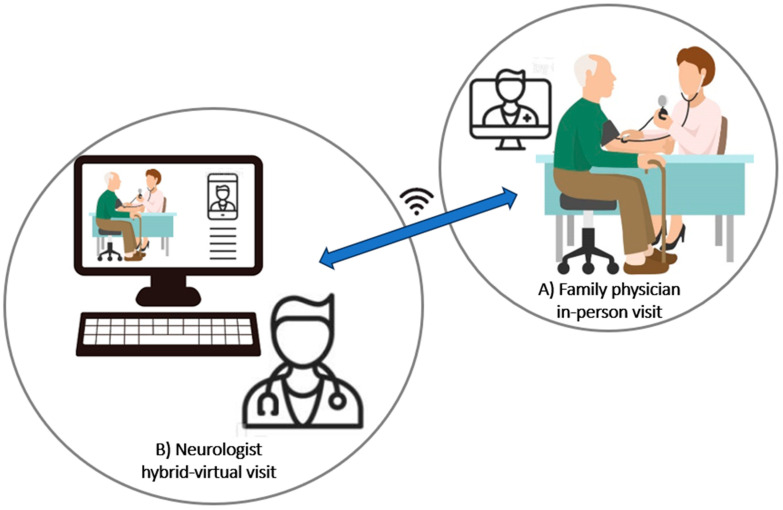
Hybrid model of virtual encounter: neurologist is virtually connected to an in-person visit, in which a patient with a neurodegenerative condition (e.g., Parkinson’s disease) can be reliably examined by a family physician to check for Parkinsonian features.

**Figure 2 healthcare-11-03187-f002:**
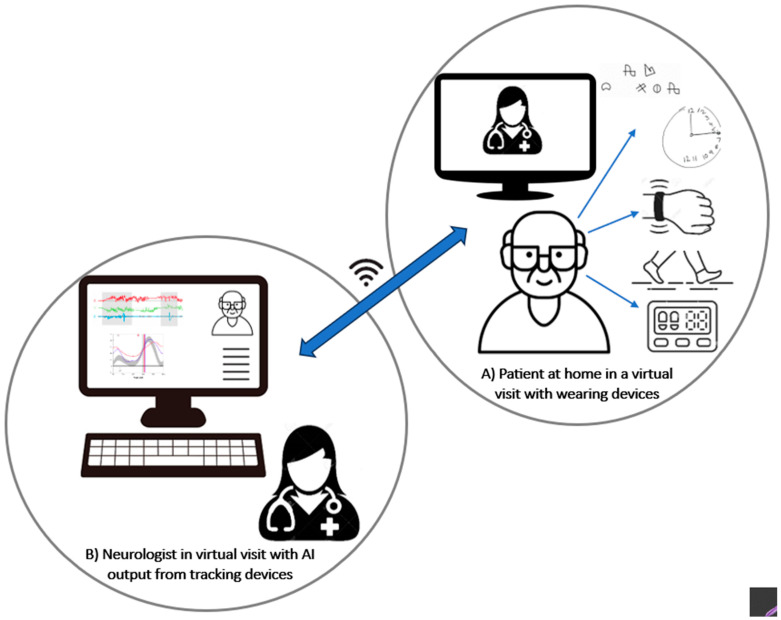
AI-assisted model of virtual encounter: neurologist is virtually connected to the patient with a neurodegenerative condition (e.g., Parkinson’s disease), who is wearing tracking devices; data are processed by AI and transferred live to the neurologist interface.

## Data Availability

Not applicable.

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
