# Peer review of "Challenges of Teleneurology in the Care of Complex Neurodegenerative Disorders: The Case of Parkinson’s Disease with Possible Solutions"

_healthcare, 2023, doi:10.3390/healthcare11243187_

Round 1

Reviewer 1 Report

Comments and Suggestions for Authors

In this work, the authors explore the field of Teleneurology, focusing on delivering neurological care virtually. While successful in acute and certain chronic conditions, they address limitations in cases like Parkinson's disease where physical examinations are crucial. To overcome these challenges, they propose two approaches: 'Hybrid Teleneurology,' involving a revised virtual encounter workflow, and 'AI-Assisted Teleneurology,' utilizing biosensors, wearables, and artificial intelligence for more accurate neurological assessments. These strategies aim to enhance the value of virtual encounters by improving the validity of neurological examinations. Although the authors have given interesting perspectives about the topic; there are a few concerns that need to be addressed to improve the quality of the work.

1. Since the structure of 'Perspective' article type follows a similar one to the ''review' article type, the authors are suggested to briefly highlight the inclusion and exclusion criteria of the selected literature to write this perspective. 

2. The authors are suggested to provide more details about the methodology used to propose and evaluate the proposed approaches ('Hybrid Teleneurology' and 'AI-Assisted Teleneurology')? How was the data collected, and what criteria were used to assess the success or feasibility of these proposed approaches?

3. In such approaches, ethical considerations are necessary. The authors are suggested to briefly mention the ethical considerations associated with the use of AI in healthcare, particularly in Teleneurology.

4. Similarly, there should be more discussion on state-of-the-art privacy or security measures taken into account in the implementation of AI-assisted Teleneurology.

5. The statements in lines 165-167 are unclear. At one point, authors mentioned that the majority of these technologies are not broadly available, how do patients already possess technologies in home settings? The authors are suggested to cross-check and improve the statement for better readability.  There should be a proper reference for the lines 171-172. 

6. In the Abstract, the authors claimed they propose these two approaches. However, such approaches are already available in my clinical settings which is not entirely true. The authors are suggested to highlight the recent advancements using device names, technology involved, and use cases to strengthen their perspective. 

7. Unlike research articles, perspectives are subjective and focus on interpreting existing literature, offering insights, and contributing to broader understanding. The work should be carefully modified to include thoughtful reflections and interpretations based on the author's expertise and experience. Therefore, the author's viewpoint on future research and development opportunities should be clearly highlighted before the conclusion section.

Author Response

Reviewer #1

In this work, the authors explore the field of Teleneurology, focusing on delivering neurological care virtually. While successful in acute and certain chronic conditions, they address limitations in cases like Parkinson's disease where physical examinations are crucial. To overcome these challenges, they propose two approaches: 'Hybrid Teleneurology,' involving a revised virtual encounter workflow, and 'AI-Assisted Teleneurology,' utilizing biosensors, wearables, and artificial intelligence for more accurate neurological assessments. These strategies aim to enhance the value of virtual encounters by improving the validity of neurological examinations. Although the authors have given interesting perspectives about the topic; there are a few concerns that need to be addressed to improve the quality of the work.

  1. Since the structure of 'Perspective' article type follows a similar one to the ‘review' article type, the authors are suggested to briefly highlight the inclusion and exclusion criteria of the selected literature to write this perspective. 

We totally appreciate this relevant comment. As per the reviewer’s suggestion, we have now added a new section into our revised article called “Methods” in which we have explained our search strategy as follows:

To select relevant articles, we performed a comprehensive exploration of the literature in the PUBMED/MEDLINE database. The primary objective was to identify publications, including original studies and review articles, focusing on teleneurology applications in the context of neurodegenerative diseases. The search utilized a combination of keywords and Medical Subject Headings (MeSH) terms, including "teleneurology," "telemedicine," "neurodegenerative diseases," "Parkinson's disease," "Alzheimer's disease," and related terms. The inclusion criteria were restricted to articles published in English, aligning with the language of the review. A systematic approach was adopted, incorporating Boolean operators to refine the search scope and ensure relevance to the objective of assessing teleneurology interventions in the context of neurodegenerative conditions. This strategy aimed to capture a comprehensive and representative body of literature to inform the narrative synthesis and discussions within the current perspective article.”

  1. The authors are suggested to provide more details about the methodology used to propose and evaluate the proposed approaches ('Hybrid Teleneurology' and 'AI-Assisted Teleneurology')? How was the data collected, and what criteria were used to assess the success or feasibility of these proposed approaches?

We appreciate this comment. Please note that this article is neither original research nor a systematic review. This article is a perspective review where we aim to provide an overview of current trends, advancements, and emerging concepts within the field of teleneurology, in particular for the care of patients with Parkinson’s disease. Also, we have recommended two new models of adjusted teleneurology to improve the quality of virtual care for these patients. These models have not been previously implemented with the exact same model; yet, there are some previous works with nearly identical models that we have summarized in our manuscript. As per the reviewer’s suggestion, we have now added new paragraphs to both sections ('Hybrid Teleneurology' and 'AI-Assisted Teleneurology') to further mention existing experience with these similar models. The additions are as follows:

“It should be noted that the term ‘hybrid teleneurology’ has also been used in a different context, where it refers to a model of care with both conventional in-person visits and remote virtual care combined. In one example of this type of so-called ‘hybrid teleneurology’, data from a neuromuscular clinic in Greece revealed equal quality of care via the ‘hybrid’ model of both tele-consultation and face-to-face in-person visits during the SARS-CoV-2 pandemic compared to conventional all-in-person visits provided during the pre-pandemic era [30].”

“AI-assisted teleneurology has been previously conceptualized in other contexts, such as telestroke, which includes using machine-learning algorithms for timely analysis of brain imaging in the acute phase of stroke, providing decision support systems to assist with treatment plans (e.g., thrombolysis, thrombectomy), and speech recognition tools, all of which can improve stroke outcomes [41].”

  1. In such approaches, ethical considerations are necessary. The authors are suggested to briefly mention the ethical considerations associated with the use of AI in healthcare, particularly in Teleneurology.

Thank you for raising this very important issue. As per the comment, we have now added a new paragraph to section ‘5. Artificial Intelligence (AI)-Assisted Teleneurology: Role of Biosensors’ to explicitly discuss the ethical issues surrounding the use of AI in telehealth. The paragraph is as follows:

“The potential for bias in AI algorithms poses challenges, as it could lead to disparities in diagnosis and treatment recommendations [45]. Most importantly, the integration of AI in teleneurology, and in healthcare in general [46], necessitates careful consideration of ethical implications. Ethical concerns surround issues such as data privacy, security, and patient consent, given the sensitive nature of neurological health information. Continuous data monitoring in healthcare, while offering valuable insights, raises legitimate privacy concerns that necessitate careful consideration of patient consent. Patients may express concerns about the extent to which their data is being monitored, shared, or stored, prompting the need for transparent and comprehensive consent processes. Adequate informed consent procedures must be in place, clearly outlining the scope of data collection, potential uses, and any sharing practices. Transparency in AI decision-making processes and engaging patients in this process becomes crucial to building trust among patients and healthcare professionals. Additionally, issues related to accountability, responsibility, and the potential for over-reliance on AI recommendations merit careful consideration. To ensure privacy or security measures, robust encryption protocols are required to secure data transmission during remote consultations. Access controls and authentication mechanisms together with regular security audits should be implemented to restrict unauthorized entry to AI systems and patient databases. The ongoing development of teleneurology presents a significant challenge in maintaining ethical norms while effectively utilizing AI to improve diagnostic accuracy and patient care in people with neurodegenerative disorders. The continuous consideration of ethical principles and the development of explicit protocols are important to effectively negotiate the intricacies involved and guarantee the conscientious implementation of AI technologies within the field of neurological healthcare.”

  1. Similarly, there should be more discussion on state-of-the-art privacy or security measures taken into account in the implementation of AI-assisted Teleneurology.

Thanks for the relevant comment. As mentioned above, we have now added a new paragraph to this section to further discuss the ethical considerations of AI systems in teleneurology. One of the aspects is privacy and security which we have covered in the following paragraph: 

“To ensure privacy or security measures, robust encryption protocols are required to secure data transmission during remote consultations. Access controls and authentication mechanisms together with regular security audits should be implemented to restrict unauthorized entry to AI systems and patient databases.”

  1. The statements in lines 165-167 are unclear. At one point, authors mentioned that the majority of these technologies are not broadly available, how do patients already possess technologies in home settings? The authors are suggested to cross-check and improve the statement for better readability. There should be a proper reference for the lines 171-172. 

Thanks for raising this issue, as per the comment, we have edited the sentence (previously aligned with lines 165-167) to clarify the concept and delete the ambiguity, as follows:

“There is a growing surge in the availability of these technologies worldwide, and many patients already possess technologies that can supplement the remote neurologic examination [42].”

We have also added a new reference (number 44) to cite the sentence in previous lines 171-172, as follows:

“A significant number of patients with neurodegenerative conditions such as PD are old by age, therefore, not technologically savvy per se; also, may develop cognitive impairments even during the prodromal stages [43] resulting in challenges using smartphones or biosensors properly [44].”

  1. In the Abstract, the authors claimed they propose these two approaches. However, such approaches are already available in my clinical settings which is not entirely true. The authors are suggested to highlight the recent advancements using device names, technology involved, and use cases to strengthen their perspective. 

We thank the reviewer for this comment. We totally appreciate the fact that teleneurology has become an established care model, and both hybrid models and AI-assisted technology have already been in use in some centers. However, what we are emphasizing are some innovative adjustments to make these models more efficient and propose solutions for the caveats of current teleneurology models, in particular the difficulties that exist with virtual neurological exams focusing on people with Parkinson’s disease. As per the reviewer’s comment, we have now revised the statement in the abstract as follows:

“Therefore, in this perspective review, we discuss two promising teleneurology approaches and propose adjustments to enhance the value of virtual encounters by improving the validity of neurological examination: the ‘hybrid teleneurology’ by revising the workflow of virtual encounters, and the ‘artificial intelligence (AI)-assisted teleneurology’ by using biosensors and wearables and data processing using AI.”   

Additionally, as per the second part of this comment, we have now added a new paragraph in section ‘5. Artificial Intelligence (AI)-Assisted Teleneurology: Role of Biosensors‘ to highlight the recent advancements in technology for monitoring symptoms in Parkinson’s disease with naming specific devices, and technology involved. Here is the new paragraph:

“A large list of sensor-based wearable devices including body attached sensors, ubiquitous networking, and embedded sensors have been applied to track motor symptoms, which serve as valuable tools in capturing baseline and longitudinal neurological data in people with PD [34]. For instance, using an inertial measurement unit -based motion capture system researchers have objectively measured full-body tremor in a cohort of individuals with PD [35]. Lightweight and wearable inertial sensors have been proposed for monitoring freezing of gait in people with PD [36]. There exist many other e-health wearables that are provided with the capability to integrate contextual data [34]; some examples are: DynaPort Hybrid system (worn on the lower back) [37], Parkinson’s Kinetigraph (a wrist worn logger) [38], SENSE-PARK system (for monitoring gait, hypokinesia, dyskinesia, sleeping) [39], and GaitRite System (for monitoring gait parameters) [40], to name a few.”

  1. Unlike research articles, perspectives are subjective and focus on interpreting existing literature, offering insights, and contributing to broader understanding. The work should be carefully modified to include thoughtful reflections and interpretations based on the author's expertise and experience. Therefore, the author's viewpoint on future research and development opportunities should be clearly highlighted before the conclusion section.

We totally appreciate this relevant recommendation. As per the reviewer’s suggestion, we have now added a new section before the conclusion, titled ‘Future Directions’ where we have explained our viewpoint on future research and development opportunities in the field of teleneurology. The section is as follows:

6. Future Directions

The future of teleneurology holds immense promise with the integration of wearable devices for data collection and the utilization of AI technologies. Wearable devices equipped with advanced sensors can offer continuous, real-time monitoring of neurological parameters, providing invaluable insights into patients' daily activities and health trends. This opens avenues for more comprehensive and personalized care, especially in managing chronic neurological conditions. The continuous streams of data from wearables present rich opportunities for research in understanding disease progression, treatment efficacy, and identifying early warning signs. Furthermore, the application of AI in teleneurology can significantly enhance diagnostic accuracy, treatment planning, and patient management. Machine learning algorithms can analyze complex neurological data patterns, aiding in early detection of abnormalities and predicting disease trajectories, particularly in the case of neurodegenerative diseases.

The future direction of 'hybrid teleneurology' is poised for innovative shifts in workflow that aim to elevate the quality of data collection and enhance the overall efficiency of remote neurological assessments. Advancements in user-friendly interfaces, coupled with intuitive data capture methods, will streamline the process of information exchange between patients and neurologists. The emphasis will be on refining the standard neurological examination components to suit remote settings, fostering a more dynamic and interactive approach. This evolution in workflow innovation seeks to overcome barriers associated with traditional telehealth encounters, ensuring that 'hybrid teleneurology' not only remains a viable alternative but emerges as a preferred and effective mode of delivering neurological care. Collaborative efforts between primary care physicians, neurologists, IT technologists, and data scientists will be essential in unlocking the full potential of teleneurology, ushering in a new era of patient-centered, data-driven neurological care. As technology continues to advance, the synergy between wearable devices and AI holds the key to transformative breakthroughs in remote neurological healthcare.”

Reviewer 2 Report

Comments and Suggestions for Authors

This paper propose  two novel approaches to enhance the value of virtual encounters by improving the validity of neurological examination, hybrid and AI-based. I totally found this article of no use at all there are no research bases, nothing significant in research, why it was conducted etc. I cannot help myself but cannot accept it in its present form. Here are my comments that might help authors in future.

1. Introduction is incomplete. Authors need to be very specific that why did they need to conduct this research? What are the targets and goals. 

2. Nothing has been achieved in this research work. We might be able to cope with the fact that there was no comparison (its should have been there) at all but there is nothing realted to the fact that what authors have achieved.

3. Everything seems to be in air, what is the research work? What are the basis of the authors proposal. I was unable to find any significant or factual fact that should be part of any research.

4. In conclusion section there is nothing evident or solid.

Comments on the Quality of English Language

Fine

Author Response

Reviewer #2

This paper propose two novel approaches to enhance the value of virtual encounters by improving the validity of neurological examination, hybrid and AI-based. I totally found this article of no use at all there are no research bases, nothing significant in research, why it was conducted etc. I cannot help myself but cannot accept it in its present form. Here are my comments that might help authors in future.

  1. Introduction is incomplete. Authors need to be very specific that why did they need to conduct this research? What are the targets and goals. 

Please note that this article is not a research paper but a perspective review. Yet, as recommended, we have now added further rationales to the ‘Introduction’ section, explaining why new innovative platforms and workflows are required to enhance the efficiency of teleneurology. We have added several new references in this paragraph. The new paragraph is as follows:

“One key challenge is the reliance on comprehensive neurological examinations, which traditionally involve hands-on assessments of motor and sensory functions, cerebellar, gait, and cognitive changes. The limitations of virtual platforms in capturing these nuances can hinder accurate diagnosis and disease monitoring. A recent scoping review of 37 studies on teleneurology highlighted the essential complementary role of traditional in-person healthcare in conjunction with telemedicine to increase the value of virtual care for neurological diseases [11]. Additionally, the elderly population affected by Alzheimer's or other dementia (including Parkinson's disease dementia) may face barriers related to technology literacy and access, posing challenges for their effective engagement in telehealth initiatives [12]. The lack of standardized protocols for remote neurological assessments and the potential for variations in the quality of internet connectivity further contribute to the challenges in ensuring consistent and reliable teleneurology services. Accelerated attempts during the SARS-CoV-2 pandemic have resulted in the development of guidelines for virtual neurological examination [13], which should be considered to improve the quality of neurological examination in teleneurology. As technology evolves and healthcare systems adapt, addressing these challenges will be essential for optimizing the efficacy of teleneurology in delivering comprehensive and patient-centered care for those with neurodegenerative diseases.”

  1. Nothing has been achieved in this research work. We might be able to cope with the fact that there was no comparison (its should have been there) at all but there is nothing related to the fact that what authors have achieved.

Please note that this article is not original research work. This article is a perspective review. A perspective review article provides an overview and analysis of current trends, advancements, and emerging concepts within a specific field of study, here the field of teleneurology care for neurodegenerative diseases like Parkinson’s disease. As such, there is no research-based data from our own projects to present here. Also, unlike traditional review articles that systematically summarize existing literature, a perspective review focuses on the author's insights, opinions, and interpretations of the field.

  1. Everything seems to be in air, what is the research work? What are the basis of the authors proposal. I was unable to find any significant or factual fact that should be part of any research.

Once again, please note that this article is not a report of an original research project. This article is a perspective review. In review articles, authors do not have research data of their own to present. As a perspective review article, we aimed to review some relevant literature on teleneurology and its shortcomings, then we suggest two innovative models to enhance the value of teleneurology. Given the reviewers’ comments, we have now added more citations and literature review summarizing the results of the relevant research projects in this field.

  1. In conclusion section there is nothing evident or solid.

We appreciate this comment. As such, we have now expanded the ‘Conclusion’ section as follows:

“While teleneurology offers unprecedented opportunities for remote patient care, our review underscores the challenges associated with virtual neurological examinations, particularly in the context of diseases like PD. In the era of teleneurology, the in-person physical examination must be supplanted by a virtual version as neurologists transition from traditional to virtual workflows. However, virtual examination is limited in value, and is not feasible for some neurologic maneuvers such as testing for rigidity in PD. Recognizing the potential for innovation, two novel models were proposed in this article to enhance the yield of virtual encounters by focusing on improving the value of neurological examination: the ‘hybrid teleneurology’ by revising the workflow of virtual encounters, and the ‘AI-assisted teleneurology’ by applying information technology tools and data processing using AI. The 'hybrid teleneurology' model redefines the workflow of virtual encounters by seamlessly integrating in-person neurological exams at the office of the referring family physician. This approach preserves the crucial hands-on assessments while leveraging the convenience of telehealth. Furthermore, the 'AI-assisted teleneurology' model introduces a paradigm shift by harnessing information technology tools and AI-driven data processing on information gathered from wearables and sensors. This not only enhances the diagnostic accuracy of virtual assessments but also enables continuous monitoring and early detection of neurological changes. As we navigate the evolving landscape of teleneurology, these innovative models hold the potential to revolutionize patient care, bridging the gap between remote consultations and comprehensive neurological examinations. Embracing these models will pave the way for a more inclusive, effective, and patient-centric approach to neurology care in the digital age.”

Additionally, we have now added a new section called, ‘Future Directions’ prior to the final Conclusions, to further clarify the messages from our review article, as well as speculating the future direction of the field. 

Reviewer 3 Report

Comments and Suggestions for Authors

Thank you very much for giving me the opportunity to review this interesting manuscript titled Challenges of Teleneurology in the Care of Complex Neurodegenerative Disorders: The case of Parkinson’s disease with possible solutions”. With some improvements, the manuscript has the potential to be useful in the field. Overall the manuscript is appropriate for the scope of the Journal. The writing style is appropriate for a scientific manuscript. Although there are some suggestions:

1.     Introduction

Add some extra information about the escalating disparity between the availability and need for neurologists holds significant importance. Additional evidence or statistics that support the severity of this gap are needed.

Regarding the impact of long wait times for neurologic assessment, consider providing more details on the consequences and potential negative effects on patient outcomes.

2.     Teleneurology: Current Situation & Challenges

It may be helpful to briefly discuss the importance of secure in telemedicine platforms. This would underscore the significance of protecting patient privacy and data security in teleneurology.

Regarding the growth of teleneurology (3.9% to 94.6%) at the Neurological Institute at Cleveland Clinic, please provide information (timeframe, contributing factors).

In pages 69-71 please provide more references about the equal effectiveness and patient satisfaction in chronic conditions such as movement disorders, epilepsy, multiple sclerosis, dementia, and headache.

3.     Hybrid Teleneurology: Revisited Virtual Care Workflow

Good effort by the authors

4.     Artificial Intelligence (AI)-Assisted Teleneurology: Role of Biosensors

Please, discuss the strengths and limitations of correlations between smartwatch-based app measurements and established clinical scales like the MDS-UPDRS.

Ethical and Privacy Considerations related to the use of biosensors and AI in teleneurology??? Discuss privacy concerns associated with continuous data monitoring and potential implications for patient consent.

Author Response

Reviewer #3

Thank you very much for giving me the opportunity to review this interesting manuscript titled “Challenges of Teleneurology in the Care of Complex Neurodegenerative Disorders: The case of Parkinson’s disease with possible solutions”. With some improvements, the manuscript has the potential to be useful in the field. Overall, the manuscript is appropriate for the scope of the Journal. The writing style is appropriate for a scientific manuscript. Although there are some suggestions:

  1. Introduction

Add some extra information about the escalating disparity between the availability and need for neurologists holds significant importance. Additional evidence or statistics that support the severity of this gap are needed.

Thanks for this very relevant comment. As per the reviewer’s suggestion, we have now added a new paragraph to mention some statistics (with their corresponding citations) on the magnitude of the mismatch and disparity between the availability and need for neurologists. The new paragraph is as follows:  

“According to the World Health Organization (WHO), there is a substantial and growing imbalance in the distribution of neurologists, with many regions facing a severe shortage. For instance, there are only three adult neurologists per 10 million people in low- and middle-income countries, whereas high-income countries have around 160 times more neurologists in proportion to their population [5]. This disparity leaves vast populations with limited access to specialized neurological care. Though of a lower magnitude, the same problem also exists in most high-income countries fueled by factors such as an aging population and the rising prevalence of neurological disorders. As per the American Academy of Neurology (AAN), by 2025, projections are that 41 states of the United States will labor under a mismatch between the need for neurologists and the availability of neurologists, and 88% of the states with mismatch over 20% [6].”

Regarding the impact of long wait times for neurologic assessment, consider providing more details on the consequences and potential negative effects on patient outcomes.

This is indeed another valid point. We have now mentioned data from another study on delayed neurological visits in England, also, another sentence is added to the Introduction section to provide more details on the consequences of delayed diagnosis and treatment in people with Parkinson’s disease. The added sentences are as follows:

“A survey conducted in England, including over 10,000 patients, revealed that individuals with neurological disorders such as PD encounter significant delays in accessing specialized healthcare services; the wait time could be over 12 months after being referred by their general practitioners in one-third of these patients [8].”

“In the case of people with PD, delayed diagnosis and treatment can deteriorate quality of life, worsen mobility and increase the risk of fall.”

  1. Teleneurology: Current Situation & Challenges

It may be helpful to briefly discuss the importance of security in telemedicine platforms. This would underscore the significance of protecting patient privacy and data security in teleneurology.

Thank you for raising this important issue. As recommended, we have now added a brief paragraph into the ‘Teleneurology: Current Situation & Challenges’ section to mention security as another challenging aspect of teleneurology. A recent corresponding reference has been cited. The new paragraph is as follows:

“Teleneurology and telemedicine in general are facing other challenges, too. Ensuring robust security measures in telemedicine platforms is paramount for safeguarding patient privacy and data integrity. As the healthcare landscape increasingly embraces digital technologies, the confidential and sensitive nature of neurologic consultations mandates a heightened focus on security. A breach in telemedicine security, which can be associated with environmental, technology, and operational factors, not only jeopardizes patient trust but also exposes individuals to the risk of unauthorized access to their health information [26].”

Regarding the growth of teleneurology (3.9% to 94.6%) at the Neurological Institute at Cleveland Clinic, please provide information (timeframe, contributing factors).

As per the reference, this remarkable change occurred only in a short time frame of only 3 weeks in March 2020. As per the comment, we have now added the following paragraph to explain the time frame and its associated factors:

“During and after the SARS-CoV-2 pandemic, there has been a tremendous rise in the use of teleneurology, for instance, from 3.9% to 94.6% at the Neurological Institute at Cleveland Clinic, USA, which occurred in a short time frame of only 3 weeks in March 2020 [21]. The ability for this rapid adaptation relies on factors such as the readiness of technological infrastructure, isolation mandates of the SARS-CoV-2 pandemic, and health information literacy of the staff.”

In pages 69-71 please provide more references about the equal effectiveness and patient satisfaction in chronic conditions such as movement disorders, epilepsy, multiple sclerosis, dementia, and headache.

As per this comment, we have now provided separate references for each of these neurological conditions, as follows:

“In recent decades, teleneurology has been used successfully in both acute care setting such as  stroke ("telestroke") [14], and outpatient teleconsultations for chronic neurologic conditions [15] such as movement disorders [16], epilepsy [17], multiple sclerosis [18], dementia [19], and headache [20].”

  1. Hybrid Teleneurology: Revisited Virtual Care Workflow. Good effort by the authors.

Thanks for your positive feedback.

  1. Artificial Intelligence (AI)-Assisted Teleneurology: Role of Biosensors

Please, discuss the strengths and limitations of correlations between smartwatch-based app measurements and established clinical scales like the MDS-UPDRS.

We appreciate this interesting comment. Accordingly, we have now added a new sentence to this section to briefly discuss the strength and limitations of these methods, as follows:

“Although data collected from these sensors can be noisy and sometimes difficult to interpret, these methods of continuous data collection have indeed some strengths. The assessments conducted in clinic frequently failed to accurately represent the patients' usual condition within their home environment; however, the wearables and sensors that are worn for long hours at home collect real-time data in real environment.”

Ethical and Privacy Considerations related to the use of biosensors and AI in teleneurology??? Discuss privacy concerns associated with continuous data monitoring and potential implications for patient consent.

Thanks for this very important comment, which is in line with one of the previous comments from Reviewer #1. As per this suggestion, we have now added extensive discussions in the ‘Artificial Intelligence (AI)-Assisted Teleneurology: Role of Biosensors’ section to cover these issues as follows:

“Most importantly, the integration of AI in teleneurology, and in healthcare in general [46], necessitates careful consideration of ethical implications. Ethical concerns surround issues such as data privacy, security, and patient consent, given the sensitive nature of neurological health information. Continuous data monitoring in healthcare, while offering valuable insights, raises legitimate privacy concerns that necessitate careful consideration of patient consent. Patients may express concerns about the extent to which their data is being monitored, shared, or stored, prompting the need for transparent and comprehensive consent processes. Adequate informed consent procedures must be in place, clearly outlining the scope of data collection, potential uses, and any sharing practices. Transparency in AI decision-making processes and engaging patients in this process becomes crucial to building trust among patients and healthcare professionals. Additionally, issues related to accountability, responsibility, and the potential for over-reliance on AI recommendations merit careful consideration. To ensure privacy or security measures, robust encryption protocols are required to secure data transmission during remote consultations. Access controls and authentication mechanisms together with regular security audits should be implemented to restrict unauthorized entry to AI systems and patient databases. The ongoing development of teleneurology presents a significant challenge in maintaining ethical norms while effectively utilizing AI to improve diagnostic accuracy and patient care in people with neurodegenerative disorders. The continuous consideration of ethical principles and the development of explicit protocols are important to effectively negotiate the intricacies involved and guarantee the conscientious implementation of AI technologies within the field of neurological healthcare.”

Round 2

Reviewer 1 Report

Comments and Suggestions for Authors

The authors have addressed most of the concerns raised by the reviewer and the manuscript can now be accepted.

Author Response

We appreciate the reviewer’s comments, which indeed resulted in a substantial improvement in the quality of our manuscript.  

Reviewer 2 Report

Comments and Suggestions for Authors

I think authors should be able to be able to compare the others work data and workflow to compare and so that novel propose approaches make more sense. 
The Introduction now at least shows what authors want but can be improved a lot. The references enters now should have been introduced in earlier versions. Conclusions can still be significantly improve. 
However, as the authors already made the possible changes as per my previous recommendations. I am accepting it with minor revision as mention above.

Comments on the Quality of English Language

English is fine.

Author Response

- I think authors should be able to compare the others work data and workflow to compare and so that novel propose approaches make more sense.

We thank the reviewer for this relevant comment. As per the suggestion, we have now added a new paragraph to the section ‘4. Hybrid Teleneurology: Revisited Virtual Care Workflow’ to explain and compare the workflow of current hybrid telehealth models with our proposed innovative model as follows:

“The workflow is, however, quite different from our proposed model of ‘hybrid teleneurology’. In traditional ‘hybrid telehealth’ models, patients interact with their physicians in two different ways: a typical virtual encounter over camera and a conventional in-person visit at the clinic, occurring either before or after the virtual encounter. In this workflow, the potential risk of an invalid neurological exam of the virtual meeting still exists. Nonetheless, in our proposed innovative ‘hybrid teleneurology’ model, the virtual encounter with the neurologist is designed to happen simultaneously with the in-person visit of the patient and primary physician (e.g., family physician), so that medically trained personnel can perform a proper neurological examination guided by the online neurologist.”

- The Introduction now at least shows what authors want but can be improved a lot. The references enters now should have been introduced in earlier versions. Conclusions can still be significantly improve. However, as the authors already made the possible changes as per my previous recommendations. I am accepting it with minor revision as mention above.

Thanks for your positive feedback on the improvement of the ‘Introduction’ section with additional references. As per the suggestion, we have now added several new paragraphs to the ‘Conclusion’ section for further improvement as follows:

“Similar technologies and models might be in practice in some centers; however, we proposed innovative modifications to these teleneurology models to improve their efficacy, particularly the value of virtual neurological examination. The 'hybrid teleneurology' model redefines the workflow of virtual encounters by seamlessly integrating in-person neurological exams at the office of the referring family physician while the neurologist is connected online simultaneously to guide the examination and live consultation.”

“There exist many biosensors and wearables that provide the capability to integrate contextual data by continuously quantifying motor and, to a lesser extent, some non-motor features of PD, with great potential to be embedded in current 'teleneurology' clinics.”
